# Community density patterns estimated by species distribution modeling: The case study of an insect virus interaction

**Stéphane Dupas**[1,2]*, **Jean-Louis Zeddam**[2,3], **Katherine Orbe**[4], **Gloria Patricia Barrera Cubillos**[5], **Laura Fernanda Villamizar**[5¤], **Patricia Mora**[2], **Jovanni Suquillo**[4‡], **Olivier Dangles**[2,6‡], **Aristóbulo Lopez-Avilla**[5‡], **Alba-Marina Cotes-Prado**[5‡], **Jean-Francois Silvain**[1]

**1** Université Paris-Saclay, CNRS, IRD, UMR Évolution, Génomes, Comportement et Écologie, Gif-sur-Yvette, France, **2** Pontificia Universidad Católica del Ecuador, Facultad de Ciencias Naturales y Biológicas, Quito, Ecuador, **3** IPME, 911 Avenue Agropolis, Montpellier, France, **4** Instituto Nacional Autónomo de Investigaciones Agropecuarias, Estaci'on experimental Santa Catalina, Panamericana Sur Km 1 via Tambillo, Ecuador, **5** Corporación Colombiana de Investigación Agropecuaria [AGROSAVIA], Mosquera, Colombia, **6** Centre d'Ecologie Fonctionnelle et Evolutive, UMR 5175, CNRS, Université de Montpellier II, Université Paul Valéry Montpellier, EPHE, IRD, Montpellier, France

‡ These authors also contributed equally to this work.
¤ Current address: AgResearch Ltd., Lincoln, New Zealand
☯ These authors contributed equally to this work.
* stephane.dupas@ird.fr

**Data availability statement:** All data files are available from the Mendeley database at the following link: https://data.mendeley.com/datasets/p99dbv75c4/1.

## Abstract

We studied the interaction between the invasive potato moth *T. solanivora* and its granulovirus *PhopGV* in the northern Andes. Host density was analyzed based on 1206 pheromone trap data from 106 sampled sites in Ecuador, Colombia and Venezuela. The prevalence of the virus was assessed at 15 sites in 3 regions in Ecuador and Colombia. Infection status was analyzed for spatial structure at different scales: storage bag, storage room, field, locality, country. Locality and storage bag explained 8% and 26%, respectively of the total variance in infection status in glm analysis. The field versus storeroom effect differed between localities. GLM species distribution models were optimized for bioclimatic variables for both insects and viruses. Predicted virus prevalence was not significantly correlated with predicted host density at sampled virus sites. Over the entire climatic range covered by the study, the correlation was R=-0.053. Of the total population insect in this range, 26% were expected to be infected based on the model. This basic method of using species distribution models to analyze average correlations between species densities can help investigate statistical relationships across a range of trophic models using existing non-sympatric data, with little or no additional sampling effort. It removes confounding time-lag effects and allows the use of data collected separately in the different species. The approach is correlative, and cannot be interpreted in terms of causality or outside the study area.

**Funding:** Programme ECOFOR Ecosystèmes Tropicaux, Projet ENTOAND 2007-2011

**Competing interests:** The authors have declared that no competing interests exist.

## Introduction

Insect viruses are expected to have important effect on their insect host populations [1]. This potential impact stimulated their use to control insect pests in agriculture [2,3]. They are usually used with an inundative rather than inoculative strategy [4]. However resident populations of insect virus may have naturally tremendous impact on insect populations, considering their prevalence and diversity [5,6]. Although, ecological data on their natural interactions and impact as natural agents are still scarce. There is little assessment of top-down or bottom-up forces or even correlation patterns in these systems. As with fungi [7], a better understanding of their epidemiology and distribution should help to improve their use in biological control of insect pests, and ensure a greater sustainability of their detrimental effects on host pest populations.

Biological control of potato tuber moth (PTM) using insect virus has a long history [8]. *Phthorimaea operculella* granulovirus or *PhopGV* (Genus: Alphabaculovirus; Family Baculoviridae) was first discovered in Australia [9]. It is now used around the world for the biological control of PTMs. Many studies have been published on the use of this virus in pest control [8–10], on its pathogenicity [10–13], its genetic variations [14–16], histopathology [17], persistence in the soil [18], or co-infection [19]. However, little information is available on the epizootiology of this virus [20–22].

*Tecia solanivora* (Lepidoptera: gelechiidae) (Povolny), the guatemaltecan potato moth, is a major pest of potato in south and central america [23]. It extended its distribution area from Guatemala, to Northern Andean region down to Ecuador between 1973 and 1996 [24–26]. Because of is invasive nature, interaction with most natural or introduced biological control agents in Northern Andes is recent.

The objective of this work was first to determine the level of *PhopGV* infection of *Tecia solanivora* in the field and then to analyze the virus prevalence in relation to pest density at different altitudes and between field and storage facilities in northern Andes (Ecuador and Colombia). Furthermore, we propose an original approach using niche models to average temporal variations of host to estimate correlation patterns for integrated pest management. This method will have to be tested with simulated data to establish its efficiency.

## Materials and methods

### Insect data

*T. solanivora* densities were assessed using pheromone-baited-traps located along altitudinal gradients of greater amplitude than for virus prevalence in Ecuador, Colombia and Venezuela. Five different surveys, representing different regions and/or time periods, were conducted out (Table 1). A total of 1216 pheromone trap samples were collected from 106 sites in the Northern Andes (Fig 1). All individuals were collected according to the guidelines of the collection framework permit No. 1466 of 2014, issued by the National Environmental Licensing Agency(ANLA) to the Colombian Corporation for Agricultural Research (AGROSAVIA).

### Virus occurrence data

*T. solanivora* larvae (517) were collected from infested tubers from 15 localities distributed along 5 altitudinal gradients, three located in Colombia from 2800 to 2940 masl Subachoque, Siachoque and Zipaquira and three in Ecuador from Montufar-Carchi and Riobamba (distribution of localities in presented in Fig 1). In these localities, the granulovirus had never been applied for biological control purposes. Tubers were collected and dissected in the laboratory. Healthy larvae and/or larvae showing signs of granulovirus disease were collected

**Table 1. Description of pheromone trap surveys used in this study.**

| Region | # of Localites | Period | Frequency | # of samples |
|---|---|---|---|---|
| **Central Ecuador** | 42 | Jan 2006–Dec 2006 | 3 weeks | 714 |
| **Southern Ecuador** | 1 | Apr 2008 | 3 days | 1 |
| **Central Ecuador** | 14 | Feb 2008–Apr 2008 | 3 weeks | 9 |
| **Northern ecuador** | 7 | Sep 2009–Oct 2009 | 1 week | 54 |
| **Colombia** | 18 | July 2008; Sep–Dec 2009 | 4 days | 96 |
| **Venezuela** | 25 | June 1984–Nov 1984 | 1 week | 330 |
| **Total** | 107 | | | 1204 |

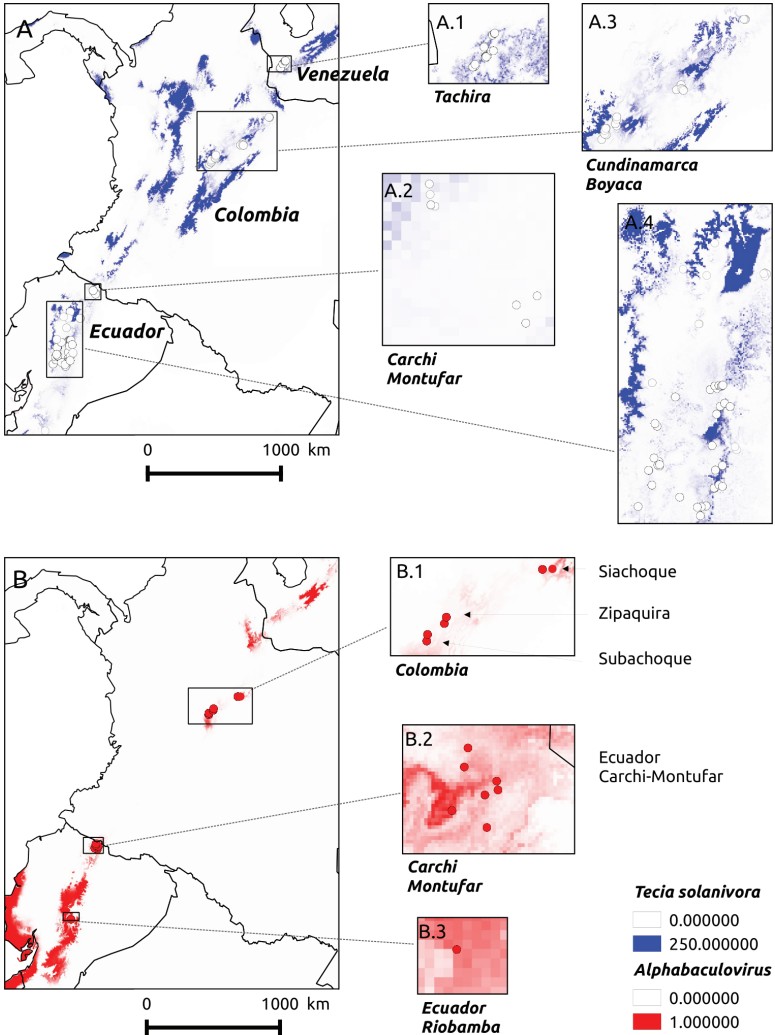

**Fig 1. Sampling sites and predicted distributions of *Tecia solanivora* and *PhopGV*.** textit*PhopGV* prevalence are estimated by PCR. (**A**) *T. solanivora* ((A.1) Venezuela, (A-2) Colombia, (A.3) Ecuador). (**B**) PhoGV ((B.1) Colombia, (B-2) Ecuador, Carchi province. (B.3) Ecuador, Riobamba province).

individually and preserved in alcohol for DNA analysis. Larval tissues were homogenized in 100 $\mu$L bidistilled $H_2O$ and centrifuged at 1000 rpm for 5 min at 4°C. To recover OBs, supernatants were centrifuged at 15000 rpm for 15 min and pellets were resuspended in 100 $\mu$L

bidistilled H$_2$O. The suspensions were incubated with 100 $\mu$L volume of 0.5 M Na$_2$CO$_3$, 50 $\mu$L SDS 10% for 10 min at 60°C and centrifuged at 5000 rpm for 5 min. The supernatants were digested with 25 $\mu$L proteinase K (20 mg/mL) at 56°C for 1 hour and DNA was extracted using a phenol/chloroform/isoamyl alcohol protocol and then precipitated with ethanol, as described in previous work [27].

The presence of the *PhopGV* was detected by PCR using primer pair, 83.2 5′-CCGCGCCG ATTACCAACAGCAGC-3′ and 84.1 5′-GAACTGTTAAACGGCTTGAGTGAGCG-3′ designed using the GenBank sequence JX170206.1. on 517 *T. solanivora* larvae collected, amplifying a 241 bp region encompassing part of the genes 83 and 84 of the *PhopGV* genome. Amplification conditions were: 94°C for 1 min, 30 cycles (94°C for 1 min, 50°C for 1 min, 72°C for 1 min), and a final extension of 7 min. Conditions were optimized separately in Colombia and Ecuador to reduce false negatives. Colombian samples were amplified in Corpoica-Tibaitata, Colombia using the following 10× mix: dNTPs 200 $\mu$M (Pharmacia Biotech 27-1850), 0.5 $\mu$M of both primers, MgCl$_2$ 2.0 mM, buffer 10× (50 mM KCl, 10 mM Tris–HCl, pH 9.0, 0.1% Nonidet), 2 U Taq polymerase (Promega M1665). Ecuadorian samples were amplified in Santa Catalina: The PCR mix contained dNTPs 250 $\mu$M (Invitrogen, reference?), 0.5 $\mu$M of both primers, MgCl$_2$ 1.6 mM, buffer 10× (50 mM KCl, 20 mM Tris–HCl, pH 8.4), 0.075 U/$\mu$L Platinum Taq polymerase (Invitrogen). All data files are available from the mendeley database PhoGV and Potato Moth geographic data in the North Andean region.

## Species distribution models

Generalized linear models (GLM) were optimized for *T. solanivora* abundances assessed by pheromone baited traps, and for *PhopGV* occurrence assessed by PCR test. Probability distributions were negative binomial with log link function for *T. solanivora* data and binomial with logit link function for *PhopGV* data. Independent variables were the region of collection (Central Ecuador, Northern Ecuador, Central Colombia, Venezuela), elevation and bioclimatic variables [28]. Bioclimatic variables were obtained by overlaying the 30″ (about 1 km) bioclimatic layers onto sites coordinates. Sites with more than 200 m difference between WorldClim and measured altitude were excluded from the analysis [29]. For the insect density the square root of altitude and all bioclimatic variables were included. For virus prevalence, only BIO1 (mean temperature), BIO6 (minimum temperature of the coldest month) and BIO12 (mean precipitation) were included. We built generalized linear models using a full backward stepwise approach using the step function of R with BIC criterion for variable selection [30]. We assumed negative binomial distribution of the response and log link function for the insect and binomial distribution of the response and logit link function for the virus. Distribution maps were reconstructed from the selected model predictions using WorldClim bioclimatic layers in a 30 arc second resolution (<1 km). Relationships between insect and virus distributions were analyzed. The grid cells of the map used for the analysis were those with all bioclimatic and elevation values within the range of the sites sampled for the virus, and for the insects. Parasite prevalence and pest density were estimated by the model in these locations. Correlation between virus prevalence and host density were plotted and calculated using GLM model analysis of deviance. Only sites where virus was sampled were considered for significance testing. Overall prevalence was calculated as the total number of infected insects divided by the total number of insects in the area considered, using the equation (1).

$$\text{Overall prevalence} = \frac{\sum_i^N prodpathogen_i Pest\ density_i}{\sum_i^N Pest\ density_i} \tag{1}$$

where $i$ and $N$ are the number of grids cells that meet the climatic conditions of the sampled area, respectively. The data were analyzed in R 3.2.2. The script is available as S1 File. The R libraries *raster*, *sp*, *mapplots* and *rgdal* were used in the script.

## Hierarchical structure of virus prevalence

Autocorrelation of virus prevalence at different scales (country, locality, site, store room/field, potato bag and tuber) was analyzed with GLM. Binomial distribution and logit link function were assumed. First model tested country and locality effects (locality nested in country). Then, nested GLM tested the effects of locality, storage bag and tuber (tuber nested in bag nested in locality). The third model tested differences between field and storage room within locality.

# Results

## Hierarchical structure

*PhopGV* prevalence was estimated using PCR detection on 517 larvae from 15 sites in Colombia and Ecuador. Important differences were observed between sites (Fig 1). The prevalence was higher in Ecuador (29%) than in Colombia (4%) (Figs 1 and 2). In Ecuador, the variation of virus infection was analyzed in relation to location, field versus potato storage rooms, potato storage bag, and potato tuber (Table 1). There were significant effects of Site ($p < 0.0001$, 53.6% of the deviance, $df = 5$) and potato bag ($p < 0.05$, 17.2% of the deviance, $df = 12$). The differences between potatoes within the bag were not significant. The differences between store and field were not significant. In the two locations where both the storerooms and the fields were sampled, the difference was not in the same direction: In one location there were significantly more virus infections in the field (Chutan bajo, Montufar, Carchi, logit +4.6, $p<0.01$) and in the other significantly fewer infections in the field (El Chamizo, Montufar, Carchi, logit –3.5, $p < 0.05$).

## Average correlations

Species distribution models were optimized for both partners using field occurrence data for the virus (517 moths at 13 sites in Ecuador and Colombia) and field abundance data for the insect (1203 pheromone trap data at 105 localities in Ecuador, Colombia and Venezuela). Stepwise GLM results are presented in S1 and S2 tables for the moth and virus, respectively. In each site sampled for *PhopGV* larval prevalencem analysis, *T. solanivora* density was estimated using the GLM regression predictor. Altitude, temperature at the driest month and average precipitation all had significant negative effect on prevalence ($p<0.0001$), precipitation (BIO12) explaining the most residual deviance (71.3%) (S2 Table). The predicted abundances of moths and viruses are shown in Fig 1. The relationship between logarithmic densities of potato moth and virus prevalence predicted by the niche models within attitudinal ranges is shown in Fig 2 for the sites where virus were sampled. Virus prevalence was not significantly dependent on host density ($R = 0.182$ *NS*, Fig 2). To have a broader view (without statistical test), we examined the relationship between all the cells of the map within the ranges of the virus localities and within the ranges of the *T. solanivora* survey localities for all the variables used in their respective models (S1 Fig). The correlation in these cells was close to zero ($R = –0.053$). The total percentage of insects infected by the virus in this climatic area was 23.6%.

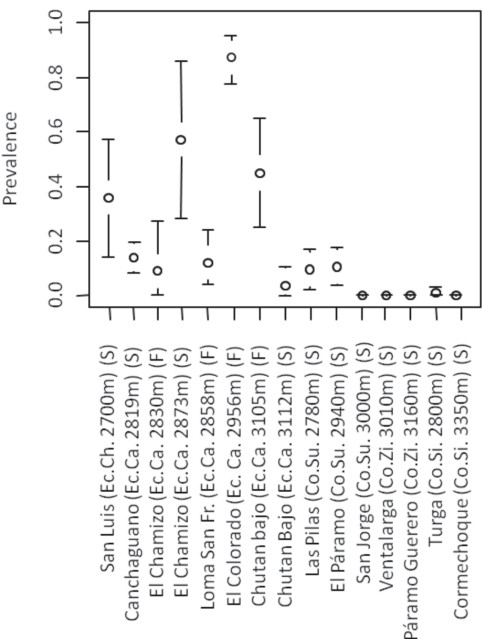

**Fig 2. PhopGV prevalence estimated by PCR.** Country, gradient and altitude are given between brackets. Co.: Colombia. Ec.: Ecuador. Ch. Chimborazo: Ca. Carchi: Su.: Subachoque, Zi.: Zipaquira, Si.: Siachoque. S: storage room sampling. F: field sampling.

## Discussion

In this study, prevalence of the PhopGV baculovirus was assessed in populations of three species of potato moth found in potato storage facilities of farmers located at different altitudes in the northern Andes. While no *PhopGV*-infected *PTM* larvae were found during the survey, the virus was relatively common among the sampled individuals from Colombia and Ecuador, but as covert infections (23.6% prevalence overall). These observations of virus prevalence in *T. solanivora* are consistent with previous reports limited to PhopGV in *P. operculella* larvae in Australia and Tunisia [20,21] and with prevalences observed up to 50% observed for granuloviruses in general [31]. In the case of PhopGV, it seems that sublethal infections seem to be the rule while symptomatic larvae are not often found. Similarly, epizootics, if any, would be exceptional.

For each elevation gradient, no significant relationship was observed between *PhopGV* prevalence and *T. solanivora* insect density (Fig 2). The lack of significant relationship between host density and virus prevalence suggests that environmental drivers of moth and virus are independent [32]. For the virus, our result suggest a significant negative effect of elevation, temperature of the driest quarter and precipitation. These results are consistent with [33] in its recent review of soil baculovirus reservoir. The negative effect of UV (related to altitude and temperature in the Andes) on virus transmission was presented as generally accepted and was again recently reported to reduce latent baculovirus transmission across crop seasons [34]. Moist soils, consistent with our rainfall effect, have also been reported to reduce baculovirus activity. Finally elevated temperatures >60°C do inactivate occlusion bodies within few minutes [33]. With more data, more factors could be included in our model, and the direct effect of climatic factors on *PhoGV* transmission could be distinguished and separated from host density effect.

We observed autocorrelations of virus prevalence at different scales (tubers, bag, storage room/field, site, and region). The observed autocorrelations suggest that bag storage contributes to *PhopGV* epizootics. Transmission of the virus may be increased by using the same bags for storage from one season to the next, and also, possibly by leaving storage residues within the room between seasons. Recycling of used bags after the tubers are sold to the market may also favor virus transmission at larger distances. In conclusion, the same factors that favor long-distance dispersal of *PTM* may also favor long distance dispersal of *PhopGV* [8,35–37].

Measuring food web patterns and processes is a major focus of ecology [32,38,39]. The characterization of species interactions is critical to predict ecosystem dynamics and stability [40]. An array of techniques have been developed to infer them from the analysis of time series [41–47]. Time lags in species response is among the confounding factors. Theory predicts ¼-period lags in consumer resource abundance, and many datasets even show consumer-resource oscillations with an antiphase (1/2-period lags) or nearly antiphase period due to eco-evolutionary dynamics [48,49]. These time lags can yield positive or negative correlations in contemporaneous samples. Inference of both time lag and interaction strength evolution relies long term time series data [50,51]. Our methodology is relatively simple and allows the measurement average density relationships between trophic levels across the geographic range of species. It does not require occurrence data to be collected at the same time and place for each species in the food web, facilitating meta-analytic approach. It should therefore be possible to use much larger databases than in the current meta-analyses which are limited to data where multiple trophic levels are examined together over large spatial and temporal scales [38]. However density estimates from niche models must be treated with caution and should not be used outside the calibration range of the model where they may not be valid [52]. In our work, we therefore removed sites outside the climatic and geographic range of the data to estimate correlations. Another issue is significance testing. A conservative test would require that the number of sites used to test for correlation not exceed the number of sites of any of the species analyzed in the test. In our work, we therefore used the sites of the species with the lowest number of sites (the virus) for correlation testing. In the context of integrated pest management strategies, habitat management variables could be added to the environmental dataset to study jointly their joint effects on pests and beneficials insects or control agents. The method is theoretically applicable to any consumer resource or food web system, and in practice should be valuable for evaluating a range biological control agents. Estimating the average density at each trophic level as a function of nutrient resource or climate can provide data for a more general view of the consequences of global changes in climate and energy and matter flows on food web structure.

## Acknowledgments

We thank all the farmers that allowed the collection of specimens in their fields.

## Supporting information

**S1 Fig. *Tecia solanivora* density and *PhoGV* prevalence, predicted by the GLM models.** Are presented in this figure only the cells of the map where predictor variables fall in the range where both *Tecia solanivora* and *PhoGV* have been sampled. (**A**) Correlation. (**B**) Points of the map represented.
(PDF)

**S1 File. R Script for the analysis of relationships between PhoGV and Potato tuber moth distributions** Bioclimatic predictor raster stacks are loaded, glm models for densities of insests and virus prevalence are fitted, and correlations between these predicted variables at the sampling points are performed. Finaly anova is performed to evaluate the hierarchical effects of site, storage bags and tubercule on virus infection. Libraries MASS, raster, mapplots, rgdal and matrixStats are used.
(R file)

**S1 Table. Stepwise polynomial generalized linear model regression procedure using BIC criteria for potato tuber moth in the north Andean region.** (A) Phthorimmaea operculella, (B) Tecia solanivora, and (C) Symetrischemma tangolias. Full model comprised following dependent variables : Alt = Alitude, BIOi = worldclim bioclimatic variables (see details below), their squared values $I(.)^2$, and the belonging to different regions: Central Ecuador (Chimborazo, Bolivar, Tungurahua, Cotopaxi, and Pichincha provinces), Northern Ecuador (Carchi), Central Colombia (Cundinamarca, Boyaca), and Venezuela. BIOi : BIO1 = Annual Mean Temperature, BIO2 = Mean Diurnal Range (Mean of monthly (max temp - min temp)). BIO3 = Isothermality (BIO2/BIO7) (* 100), BIO4 = Temperature Seasonality (standard deviation *100), BIO5 = Max Temperature of Warmest Month, BIO6 = Min Temperature of Coldest Month, BIO7 = Temperature Annual Range (BIO5-BIO6), BIO8 = Mean Temperature of Wettest Quarter, BIO9 = Mean Temperature of Driest Quarter, BIO10 = Mean Temperature of Warmest Quarter, BIO11 = Mean Temperature of Coldest Quarter, BIO12 = Annual Precipitation, BIO13 = Precipitation of Wettest Month, BIO14 = Precipitation of Driest Month, BIO15 = Precipitation Seasonality (Coefficient of Variation), BIO16 = Precipitation of Wettest Quarter, BIO17 = Precipitation of Driest Quarter, BIO18 = Precipitation of Warmest Quarter, BIO19 = Precipitation of Coldest.
(DOCX)

**S2 Table. Stepwise polynomial generalized linear model regression for pohGV prevalence in Tecia solanivora.** Alt = Alitude, BIO9 = Mean temperature of the driest quarter, BIO12 = Annual precipitation. . Alt = Alitude, BIO9 = Mean Temperature of Driest Quarter, BIO12 = Annual Precipitation.
(DOCX)

## Acknowledgments

We thank the farmers who allowed the collection of samples in their fields.

## Author contributions

**Conceptualization:** Stéphane Dupas.

**Data curation:** Stéphane Dupas, Jean-Louis Zeddam, Katherine Orbe, Gloria Patricia Barrera Cubillos, Patricia Mora.

**Funding acquisition:** Jean-Francois Silvain.

**Investigation:** Katherine Orbe, Gloria Patricia Barrera Cubillos, Laura Fernanda Villamizar, Patricia Mora, Jovanni Suquillo.

**Methodology:** Jean-Louis Zeddam.

**Project administration:** Stéphane Dupas, Aristóbulo Lopez-Avilla.

**Supervision:** Stéphane Dupas, Laura Fernanda Villamizar, Jovanni Suquillo, Aristóbulo Lopez-Avilla, Alba-Marina Cotes-Prado.

**Writing – original draft:** Stéphane Dupas.

**Writing – review & editing:** Stéphane Dupas, Jean-Louis Zeddam, Olivier Dangles, Alba-Marina Cotes-Prado, Jean-Francois Silvain.

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
