## [Decision Letter · Decision Letter 0]

6 May 2024

PONE-D-24-05136Community density patterns estimated by species distribution modeling: the case study of an insect virus interactionPLOS ONE

Dear Dr. Dupas,

Thank you for submitting your manuscript to PLOS ONE. After careful consideration, we feel that it has merit but does not fully meet PLOS ONE’s publication criteria as it currently stands. Therefore, we invite you to submit a revised version of the manuscript that addresses the points raised during the review process.

We look forward to receiving your revised manuscript.

Kind regards,

Clement Ameh Yaro, Ph.D

Academic Editor

PLOS ONE

Journal Requirements:

"Programme ECOFOR Ecosystèmes Tropicaux, Projet ENTOAND 2007-2011"

"This work has been founded by ECOFOR program ENTOAND 2007-2011"

"Programme ECOFOR Ecosystèmes Tropicaux, Projet ENTOAND 2007-2011"

Reviewers' comments:

Reviewer's Responses to Questions

**Comments to the Author**

1. Is the manuscript technically sound, and do the data support the conclusions?

Reviewer #1: Partly

Reviewer #2: Yes

2. Has the statistical analysis been performed appropriately and rigorously? 

Reviewer #1: I Don't Know

Reviewer #2: Yes

3. Have the authors made all data underlying the findings in their manuscript fully available?

Reviewer #1: Yes

Reviewer #2: Yes

4. Is the manuscript presented in an intelligible fashion and written in standard English?

Reviewer #1: Yes

Reviewer #2: Yes

5. Review Comments to the Author

Reviewer #1: This is an interesting study demonstrating a methodological framework to study patterns of pathogen prevalence and host density. However, it requires much additional work before being acceptable for publication.

My main objection lies in the apparent “hybrid” nature of the study. Whereas the introduction is mostly about developing a new methodological approach applied to a particular case, the rest of the paper concerns the specifics of the system under study.

I would challenge the purportedly methodological nature of the paper. Developing a new approach would require (i) to test the method with simulated data to establish its efficiency and (ii) to compare its results with alternative methods to assess is performance. These steps have not been undertaken in the study.

My suggestion is that the authors focus in the analysis of their specific virus-insect system and indicate somewhere in the discussion that their approach might be valuable given the difficult applicability of other approaches.

Specific comments

Abstract 5.- It is not clearly shown what this results means biologically.

51. insert “on” between “relies” and “long”.

53-54. “Casuality is still not inferred…” It is not entirely clear what is meant here.

196. “precipitation being the most significant…” This can be misleading because p-values do not indicate of the size of the effect. A very low p-value suggests a clear difference but not necessarily a large one. So being “the most significant” is not particularly informative in terms of biology.

I could not find any captions for figures 1 and 2. Apparently that of S1 Fig. corresponds actually to Fig. 1, but then what is the S1 figure included in supplementary information? Likewise, S2_Fig.pdf is not referred to in the text and the S1 R script is identified as S2 in line 285.

Reviewer #2: From my point of view, the document is well structured and supported by the data presented. This can be considered an interesting contribution to a very little studied subject in the neotropical region and particularly in countries such as Ecuador.

6. PLOS authors have the option to publish the peer review history of their article (what does this mean?). If published, this will include your full peer review and any attached files.

Reviewer #1: No

Reviewer #2: No

---

## [Author Response · Author response to Decision Letter 1]

13 Feb 2025

Editor :

RESPONSE: We included our updated statements in our cover letter

We look forward to receiving your revised manuscript.

Kind regards,

Clement Ameh Yaro, Ph.D

Academic Editor

PLOS ONE

Journal Requirements:

RESPONSE: We ensured file naming and style templates in this revision

RESPONSE: We reworked the manustript using a PLOS latex template

RESPONSE: We added this information

RESPONSE: Done

"Programme ECOFOR Ecosystèmes Tropicaux, Projet ENTOAND 2007-2011"

RESPONSE: Ok The funders had no role in study designwe mentioned this in the cover letter.

"This work has been founded by ECOFOR program ENTOAND 2007-2011"

"Programme ECOFOR Ecosystèmes Tropicaux, Projet ENTOAND 2007-2011"

RESPONSE: Ok

Reviewer #1: This is an interesting study demonstrating a methodological framework to study patterns of pathogen prevalence and host density. However, it requires much additional work before being acceptable for publication.

My main objection lies in the apparent “hybrid” nature of the study. Whereas the introduction is mostly about developing a new methodological approach applied to a particular case, the rest of the paper concerns the specifics of the system under study.

I would challenge the purportedly methodological nature of the paper. Developing a new approach would require (i) to test the method with simulated data to establish its efficiency and (ii) to compare its results with alternative methods to assess is performance. These steps have not been undertaken in the study.

My suggestion is that the authors focus in the analysis of their specific virus-insect system and indicate somewhere in the discussion that their approach might be valuable given the difficult applicability of other approaches.

RESPONSE:

We revisited the paper in order to focuss on the analysis of our specific system. The manuscrit was introduced on the question of the lack of data regarding the natural impact of viruses on insect pests despite their use in biological control. All the manustript was carefully reviewed for the english writing with machine learning.

Specific comments

Abstract 5.- It is not clearly shown what this results means biologically.

51. insert “on” between “relies” and “long”.

RESPONSE: done

53-54. “Casuality is still not inferred…” It is not entirely clear what is meant here.

RESPONSE: In the present version, the work was no longer introduced from the fundamental question of characterizing species interactions from field or experimental data, but from the more applied question of the natural impact of insect viruses on agricultural pests. In this new context, the question of causality versus correlation has been removed.

196. “precipitation being the most significant…” This can be misleading because p-values do not indicate of the size of the effect. A very low p-value suggests a clear difference but not necessarily a large one. So being “the most significant” is not particularly informative in terms of biology.

RESPONSE: It is true. We analysed percentage of deviance explained and changed to “precipitation explained the most residual deviance (71.3%, included in the S2 Table )”

I could not find any captions for figures 1 and 2. Apparently that of S1 Fig. corresponds actually to Fig. 1, but then what is the S1 figure included in supplementary information? Likewise, S2_Fig.pdf is not referred to in the text and the S1 R script is identified as S2 in line 285.

RESPONSE: Thank you. The fig. 1 and 2 captions were included in the text, and S1 and S2 Fig captions at the end of the manuscrit as requested in Plos One formating guidelines.

Reviewer #2: From my point of view, the document is well structured and supported by the data presented. This can be considered an interesting contribution to a very little studied subject in the neotropical region and particularly in countries such as Ecuador.

6. PLOS authors have the option to publish the peer review history of their article (what does this mean?). If published, this will include your full peer review and any attached files.

---

## [Decision Letter · Decision Letter 1]

17 Mar 2025

Community density patterns estimated by species distribution modeling: the case study of an insect virus interaction

PONE-D-24-05136R1

Dear Dr. Dupas,

We’re pleased to inform you that your manuscript has been judged scientifically suitable for publication and will be formally accepted for publication once it meets all outstanding technical requirements.

Kind regards,

Clement Ameh Yaro, Ph.D

Academic Editor

PLOS ONE

Additional Editor Comments (optional):

Reviewers' comments:

Reviewer's Responses to Questions

**Comments to the Author**

1. If the authors have adequately addressed your comments raised in a previous round of review and you feel that this manuscript is now acceptable for publication, you may indicate that here to bypass the “Comments to the Author” section, enter your conflict of interest statement in the “Confidential to Editor” section, and submit your "Accept" recommendation.

Reviewer #1: All comments have been addressed

Reviewer #2: All comments have been addressed

2. Is the manuscript technically sound, and do the data support the conclusions?

Reviewer #1: Yes

Reviewer #2: Yes

3. Has the statistical analysis been performed appropriately and rigorously? 

Reviewer #1: Yes

Reviewer #2: Yes

4. Have the authors made all data underlying the findings in their manuscript fully available?

Reviewer #1: Yes

Reviewer #2: Yes

5. Is the manuscript presented in an intelligible fashion and written in standard English?

Reviewer #1: Yes

Reviewer #2: Yes

6. Review Comments to the Author

Reviewer #1: This is the second version of a manuscript I previously reviewed. I agree with the new focus the authors have adopted and am satisfied with the revisions made in response to my feedback.

I only have a few minor editorial suggestions:

30. Capitalize “south and central america”.

56. For consistence, indicate the height above sea level for the Ecuadorian localities.

77-82. “M” is italicized in some instances but not in others.

100. Correct “¡ 1 km”.

109-111. Need to explain the terms in the formula: prodpathogen, Pestdesnity.

139. Delete “m” in “prevalencem”.

142. Better “explained the largest proportion of”.

161. Insert space “21]and”.

175. Correct “¿60º”

186. The period “.” is misplaced.

199. Insert “a” between “facilitating” and “meta-analytic”.

218-253. Ackowledgments are duplicated.

224. Capitalize “glm”

Reviewer #2: I have reviewed the document and have no additional comments, from the beginning I have considered it to be a well done job. In addition, I have read the responses to the comments of other reviewers and it seems to me that the changes proposed by the authors resolve the necessary elements and strengthen the final version of the document.

7. PLOS authors have the option to publish the peer review history of their article (what does this mean?). If published, this will include your full peer review and any attached files.

Reviewer #1: No

Reviewer #2: No

---

## [Editor Report · Acceptance letter]

PONE-D-24-05136R1

PLOS ONE

Dear Dr. Dupas,

I'm pleased to inform you that your manuscript has been deemed suitable for publication in PLOS ONE. Congratulations! Your manuscript is now being handed over to our production team.

Kind regards,

on behalf of

Dr. Clement Ameh Yaro

Academic Editor

PLOS ONE